# Protection of Patient Data in Digital Oral and General Health Care: A Scoping Review with Respect to the Current Regulations

Olga Di Fede [1,†] , Gaetano La Mantia [1,2,3,*,†] , Mario G. C. A. Cimino [4] and Giuseppina Campisi [1,2]

1   Department Di.Chir.On.S., University of Palermo, 90127 Palermo, Italy
2   Unit of Oral Medicine and Dentistry for Fragile Patients, Department of Rehabilitation, Fragility, and Continuity of Care, University Hospital Palermo, 90127 Palermo, Italy
3   Department of Biomedical and Dental Sciences and Morphofunctional Imaging, University of Messina, 98147 Messina, Italy
4   Department of Information Engineering, University of Pisa, 56122 Pisa, Italy
*   Correspondence: gaetano.lamantia@community.unipa.it
†   These authors contributed equally to this work.

**Abstract:** The use of digital health technologies, including telemedicine and teledentistry, has become a necessity in healthcare due to the SARS-CoV-19 pandemic. These technologies allow for the reduction of the workload of healthcare providers and the improvement of patient outcomes in cases of remote monitoring, diagnosis, and communication. While there are no doubtful benefits, there are some counterparts, such as concerns about clinical risks, data security, and privacy protection. This paper aims to review the regulations regarding the use of digital health apps and software in healthcare. This scoping review followed the PRISMA-ScR guidelines and the five-step framework of Arksey and O'Malley. Study selection was based on eligibility criteria that were defined using the population-exposure framework. The review of the articles selected (*n* = 24) found that the majority focused on data security policies in the healthcare industry, highlighting the need for comprehensive regulations and app control systems to protect patient data. The articles also emphasized the requirement for more appropriate research and policy initiatives to improve data security practices and better address privacy and safety challenges related to health-related apps. The review recognized that papers did not report consistent standards in professional obligation and informed consent in online medical consultations, with potential risks for data privacy, medical liabilities, and ethical issues. Digital health has already revolutionized medical service delivery through technology but faces some challenges, including the lack of standardized protocols for handling sensitive patient data and the absence of common legislative provisions, raising concerns about confidentiality and security. To address these issues and deficiencies, regulatory compliance is crucial to clarify and harmonize regulations and provide guidelines for doctors and the health system. In conclusion, regulating patient data, clarifying provisions, and addressing informed patients are critical and urgent steps in maximizing usage and successful implementation of telemedicine.

**Keywords:** social app; WhatsApp; GDPR; HIPAA; sensitive data; mobile health; secure messaging app; COVID; dentistry





## 1. Introduction

A rapid transition to digital health technologies is becoming effective in all medical fields [1], and e-health is considered a solution to safely provide care to patients and ensure continuous health care even at a distance [2]. The SARS-CoV-19 pandemic of recent years has accelerated all aspects of health care, including oral health care [3]. In particular, conventional dentistry was reduced and, in some cases, interrupted to minimize the risk of exposure to the SARS-CoV-19 virus for both practitioners and patients by avoiding

in-person visits. Health and dental care organizations had to propose and plan alternative protocols [1,4–6].

Telemedicine and teledentistry are used to support clinicians and patients by providing remote monitoring services, remote diagnosis, counseling, home care, and education with self-care management [7–9]; at the same time, they are useful in reducing the workload of health care providers, simplifying interprofessional communication, providing an easy way to share patient information, and giving remote instructions [10–12].

Telemedicine has revolutionized healthcare by providing patients with convenient and accessible medical services. With the help of wireless patient monitoring devices, smartphones, personal digital assistants, and tablets, patients can now connect with specialists in real-time [13]. This eliminates the need for physical visits to the hospital, reducing lost time and allowing for quicker diagnoses. The feedback system in telemedicine also allows for continuous monitoring of the patient's health status, enabling healthcare providers to track any changes and respond promptly. This proactive approach to health management aims to improve outcomes and reduce health risks.

Moreover, telemedicine promotes informed decision-making by giving patients access to their health data and enabling them to actively participate in their own care. This can result in increased patient engagement and improved health outcomes as patients are empowered to make informed choices about their health and take action to maintain good health [9,14,15].

Social media platforms, including WhatsApp, have become an integral part of modern life, with almost half of the world's population using them [16]. As a result, healthcare professionals have adopted these platforms in their daily work to communicate and share information with their peers and patients [17–22]. While there is some evidence to suggest that using social media in healthcare can have many benefits, such as improved communication and data transfer, there are also concerns about the risks associated with its use [21,23–25].

One of the main risks is the potential for breaches of patient privacy and confidentiality. Social media platforms are public forums, and patient information shared on these platforms can be easily accessed by individuals who are not authorized to view it. In addition, patients may inadvertently disclose sensitive health information on social media, compromising their privacy and putting them at risk for discrimination or other negative consequences [26].

There is a lack of consensus in the scientific literature regarding the use of social media in healthcare, with some studies highlighting its positive aspects while others focus on the negative consequences, including clinical risks to patients, data security, and privacy protection. In addition, the use of generic apps and software to exchange health data indiscriminately is not allowed, as it poses a threat to patient safety and data security [27–32]. Given these conflicting views, this paper aims to perform a scoping review of the existing literature on regulations and guidelines for telemedicine apps and software and their use among patients and specialists [33,34]. The goal of this review is to analyze all papers on the current regulatory state and issues of telemedicine and identify crucial points to be solved by further research and development.

## 2. Materials and Methods

A review of the recent literature (years 2020–2022) was conducted, focusing on telemedicine apps and software for health care and the current regulations of digital health data. This initial search revealed 190 publications from January 2020 to December 2022. This scoping review followed Preferred Reporting Items for Systematic Reviews and Meta-Analysis-Scoping Review (PRISMA-ScR) guidelines and Arksey and O'Malley's five-stage framework to identify available evidence [33]. Five iterative stages were involved in the review: (i) Identifying the research question, (ii) identifying relevant studies, (iii) selecting relevant studies, (iv) charting the data, and (v) summarizing results.

### 2.1. Eligibility Criteria

We included studies that took place between January 2020 and December 2022 and were solely focused on the use of mobile applications and software in healthcare. The review included all aspects of healthcare, including dental, nursing, and rehabilitation.

### 2.2. Study Selection

Studies were identified by electronic searches of scientific articles from different biomedical databases (i.e., Scopus, PubMed, and Medline). To minimize biases, publications were examined individually by two reviewers (GLM and ODF).

### 2.3. Search Strategy

The following search terms were used separately and in combination: social app, teledentistry, telehealth, telemedicine, privacy, policy, legacy issues, liability issues, using medical subject headings, and free text. The full-text screening was performed only by the first author, as is common when scoping reviews are conducted [34]. Our screening procedure was guided by defined inclusion and exclusion criteria developed using the population-exposure framework (PEO) (Table 1). Any disagreements were handled by a mutual conversation among authors. Duplicate papers were deleted, after which there was further scrutiny in order to assess their eligibility.

**Table 1.** Population-Exposure Framework (PEO).

| | Inclusion | Exclusion |
|---|---|---|
| Population | - Mobile apps and software used in healthcare for telemedicine services<br>- All medical fields | - Mobile apps and software used in health care for research and evaluation and the continuing education of health care providers |
| Exposure | - Studies focusing on the regulations, privacy policy, data security, and legacy issues of telemedicine services | - Studies not concerned with regulations, privacy policies, data security, or legacy issues of telemedicine services |
| Outcome | - Studies reporting on the use of social apps and software for telemedicine and related regulation | |
| Time | - Published from January 2020 to December 2022. | |
| Study type | - Primary, peer-reviewed research<br>- Full text available | |
| Language | - English | - Languages other than English |

### 3. Results

The review was conducted in accordance with the Preferred Reporting Items for Systematic Reviews and the Meta-Analysis-Scoping Review (PRISMA-ScR) guidelines to ensure a transparent and comprehensive evaluation of the available literature (Figure 1).

The results of this analysis suggest that a total of 290 records were identified in the databases (Table 2). No duplicate records were removed prior to screening, indicating that the database search was thorough. However, 45 records were flagged as ineligible by the automation tools, indicating that some records did not meet the initial inclusion criteria. After screening, the number of records was reduced to 245. This reduction was likely due to the application of additional inclusion and exclusion criteria that were more specific than the original criteria. In addition, 55 records were excluded, indicating that some records did not meet the additional inclusion and exclusion criteria. Of the 190 records that were screened for eligibility, some reports were excluded for reasons such as the unavailability

of full text or being written in a non-English language. These exclusion criteria have been established to ensure that the studies are easily accessible and can be effectively reviewed by the research team. After conducting a systematic search of multiple databases, we included 24 articles out of 290 for analysis.

**Table 2.** Summary of 24 studies, from 2020 to 2022, regarding the protection of patient data in Digital Health.

| Author (Year) | Country | Design of Study | Issues |
|---|---|---|---|
| Agarwal et al., 2020 [35] | India | Research article | Data security policies<br>Privacy policies |
| Benjumea et al., 2020 [36] | Spain | Review article | Data security policies<br>Privacy policies |
| Caetano et al., 2020 [37] | Brasil | Research article | Data security policies<br>Privacy policies |
| Ghosh et al., 2020 [38] | India | Research article | Data security policies<br>Privacy policies<br>Legacy liabilities |
| Kaplan 2020b [39] | USA | Review article | Data security policies |
| Mahtta et al., 2021 [40] | USA | Research article | Data security policies |
| Kichloo et al., 2020 [41] | USA | Review article | Data security policies |
| Moura et al., 2020 [42] | Portugal | Research article | Data security policies |
| Gowda et al., 2021 [43] | USA | Research article | Data security policies |
| Hoaglin et al., 2021 [44] | USA | Research article | Data security policies<br>Legacy liabilities |
| Pool et al., 2021 [45] | Australia | Review article | Data security policies<br>Privacy policies<br>Legacy liabilities |
| Tangari et al., 2021 [46] | Australia | Research article | Data security policies<br>Privacy policies |
| Alfawzan et al., 2022 [47] | Zurich | Review article | Data security policies<br>Privacy policies |
| Essén et al., 2022 [48] | Sweden | Research article | Data security policies<br>Privacy policies<br>Legacy liabilities |
| Grundy 2022 [49] | Canada | Review article | Data security policies<br>Privacy policies<br>Legacy liabilities |
| Maaß et al., 2022 [15] | Germany | Overview article | Data security policies<br>Privacy policies<br>Legacy liabilities |
| Mazzuca et al., 2022 [50] | Italy | Review article | Data security policies<br>Privacy policies<br>Legacy liabilities |
| Sujarwoto et al., 2022 [51] | Indonesia | Overview article | Data security policies<br>Privacy policies |
| Venkatesh et al., 2022 [52] | India | Research article | Data security policies<br>Privacy policies |
| Eisenstein et al., 2020 [53] | Brasil | Review article | Privacy policies |
| Perez-Noboa et al., 2021 [54] | Ecuador | Research article | Privacy policies |
| Wang et al., 2020 [55] | USA | Research article | Legacy liabilities |
| Lee et al., 2021 [56] | China | Research article | Legacy liabilities |
| Ferorelli et al., 2022 [57] | Italia | Review article | Legacy liabilities |

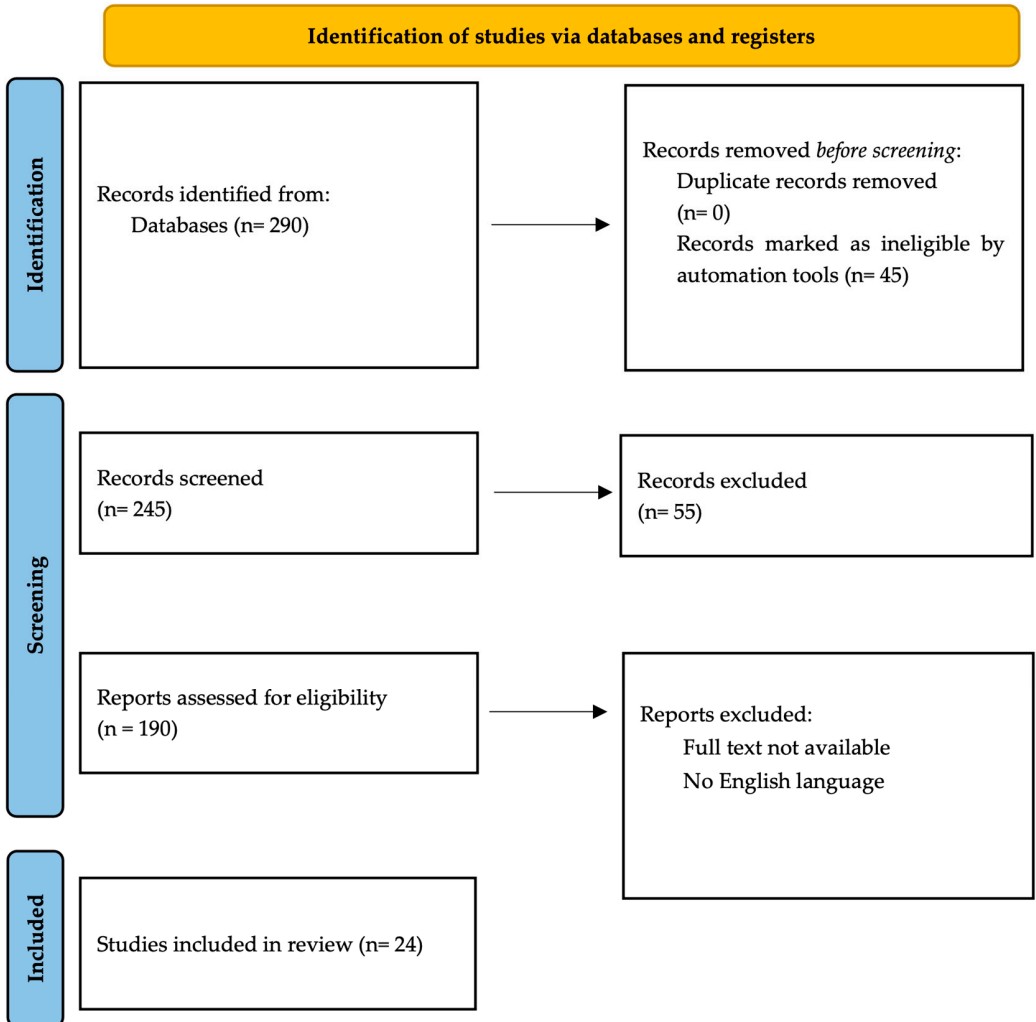

**Figure 1.** PRISMA flow diagram for systematic reviews.

Through the analysis of these 24 articles, three key themes emerged as crucial components of health app regulations: privacy policies, data security policies, and legacy liabilities. Privacy policies focus on the protection of personal information and sensitive data collected and stored by health apps, while data security policies address the measures in place to secure this information from unauthorized accesses and breaches. Legacy liabilities pertain to the legal responsibilities and obligations of health apps, particularly regarding medical information and advice provided to users. These findings highlight the importance of considering privacy, data security, and legacy liabilities in the regulation of digital health, which is rapidly becoming a popular tool for managing health and wellness.

### 3.1. Data Security Policies

Data security is a critical issue in the healthcare industry. The need for comprehensive policies and regulations is well documented in the literature. A review of 24 articles revealed that 19 [15,35–45,47–52,58] of them focused on the topic of data security policies, with a specific emphasis on the challenges faced in protecting patient data. The articles pointed out that the lack of comprehensive regulation was a major concern and that the need for an app control system was crucial in preventing the sharing of patient data with unauthorized third parties.

The majority of the articles also emphasized the need for more research in this area, particularly to address the growing concerns around data privacy and security. They suggested that awareness and initiatives by healthcare professionals, healthcare users, and decision-

makers were essential in promoting better data security practices. This is especially true in European and American nations that follow GDPR and HIPAA regulations, respectively.

Overall, the findings of these articles highlight the importance of ensuring that patient data is protected and that appropriate policies and regulations are in place to prevent unauthorized access to and sharing of sensitive information. The need for further research, awareness, and initiatives by various stakeholders in the healthcare industry cannot be overemphasized, as data security is critical to protecting the privacy and well-being of patients.

### 3.2. Privacy Policies

Health-related app policies have been widely discussed in the academic literature, with 15 [15,35–38,45–54] out of 24 articles specifically covering this topic. The studies analyzed the policies in several countries, such as India, Spain, Australia, Zurich, Canada, Brazil, Sweden, Germany, and Italy, and identified challenges related to safety and privacy. These challenges highlight the need for more robust regulations in the areas of operationalization, implementation, and international transferability of approvals.

The lack of proper regulations has been identified as a significant barrier to the widespread adoption and safe utilization of telemedicine platforms, despite their potential benefits, such as improving access to health services and reducing healthcare costs.

The studies suggest that more work is needed in the area of health-related app policies to ensure that telemedicine platforms can be effectively and safely used by people around the world. The regulation of telemedicine is a complex issue that requires cross-national collaboration and commitment to ensuring that these technologies are used to improve health outcomes.

### 3.3. Professional Legacy Liabilities

The study of 24 articles on online medical consultations revealed that 10 articles [15,38,44,45,48–50,55–57] addressed the issue of liability and legacy concerns. These articles highlighted the risks associated with informed consent, data privacy, medical negligence, and ethical issues in the context of virtual consultations.

The articles discussed the potential risks and challenges associated with providing medical services through online consultations. One of the major issues identified was the need for clear and consistent standards regarding professional liability for medical practitioners who offer online consultations. This is important as the liability issues that arise from online consultations may be different from those associated with traditional face-to-face consultations.

Informed consent was also a weakness in online consultations, as patients may not fully understand the risks and limitations of online medical services. The security of health data was also a concern, as the transmission of sensitive information over the internet could result in data breaches or unauthorized accesses. Medical negligence was another issue raised in the articles selected. The risk of medical malpractice in online consultations is significant, as medical practitioners may not be able to accurately diagnose or treat patients without physically examining them. There is also a potential for ethical issues to arise in online consultations, such as the confidentiality of medical information and the autonomy of patients. Therefore, the articles emphasized the importance of harmonizing the different laws and regulations across different jurisdictions in order to establish a uniform standard for professional liability in online consultations. This will ensure that medical practitioners are held accountable for their actions and that patients are protected from potential harm.

## 4. Discussion

The review of current regulations on patient data protection in digital oral and general health found several outcomes in privacy policies, data security, and legacy liabilities, in particular, the lack of comprehensive regulations in Europe and America. To improve digital health, there is a need to build secure and adaptable access control models. Awareness

should be raised among users, clinicians, developers, and policy makers to carefully consider the benefits and security issues of digital health. Updating guidelines for the ethical use of telemedicine is also necessary to optimize its use and ensure evidence-based practices.

Digital health reduces health disparities and improves access to care through remote screening, treatment, and monitoring. Advanced countries that are investing in digital health include the US, UK, Singapore, South Korea, Sweden, Japan, Australia, Canada, and Germany. To maximize impact, security concerns must be addressed through robust access control models that are widely used in technology. All stakeholders, including users, healthcare providers, tech developers, and policymakers, must consider the security and benefits of mHealth apps.

To enhance its impact, it could be decisive to develop flexible, robust, and risk-conscious access control models for widely used technologies. All stakeholders, including app users, healthcare providers, technology developers, and policymakers, should acknowledge and address security concerns while considering the benefits of mobile health (mHealth) apps [39,44,48,50,51,54].

The review highlights the need for enhancing healthcare providers' care services and raising public awareness on digital health to optimize its benefits. Additionally, updated guidelines for the ethical use of telemedicine and telehealth are required for physicians and organizations [41,52].

Protecting patient data in digital health care is a crucial challenge, since sensitive personal information must remain confidential and secure at all times. Regulations such as the Health Insurance Portability and Accountability Act (HIPAA) in the US and the General Data Protection Regulation (GDPR) in the EU define standards for collecting, storing, and utilizing patient data in digital healthcare [45,59].

Health care providers and organizations must implement appropriate technical and organizational measures, such as encryption and secure backups, and undergo regular security audits to secure patient data from unauthorized access, disclosure, alteration, and destruction as mandated by these regulations [60]. Telemedicine providers must also obtain patient consent for the collection and use of their data and inform patients of their rights under the GDPR and HIPAA. Additionally, patients have the right to access, correct, and delete their personal data and also have to provide their consent for their data to be used for specific purposes. Compliance with these regulations is essential for maintaining the trust of patients and ensuring the responsible and ethical use of digital health data [58].

Telemedicine, like any other form of electronic communication, is subjected to data security issues such as hacking, data breaches, and unauthorized access to patient information. To protect patient data, telemedicine providers should use secure communication methods, such as encrypted messaging and video conferencing, and comply with relevant regulations. Additionally, providers should regularly update their security measures and train staff on best practices for protecting patient data [61]. Telemedicine is a rapidly growing field with a lot of promise for improving healthcare access and outcomes. However, it is crucial for healthcare professionals to play a central role in ensuring that telemedicine visits are conducted properly and that the technology used respects patient privacy and provides high-quality care [62]. To achieve this, a comprehensive security monitoring policy is necessary. This policy should not only identify vulnerable and suspicious code but also encourage developers to adopt strong defenses against potential hacking and cloning activities [33].

At the same time, it is important for providers to use native digital health software and apps, as they are more likely to meet regulatory requirements and be more secure. Unfortunately, the most commonly used apps, such as WhatsApp, Skype, and Zoom, do not fully comply with telemedicine requirements, as reported by various studies. This highlights the need for more robust and secure telemedicine solutions [63–68].

A study that was performed for over 20,000 health-related smartphone apps, found that a significant number of these apps could potentially access and share personal infor-

mation, such as email addresses and geolocation data. This raises serious privacy concerns and highlights the need for more stringent data protection policies and security measures. In conclusion, while telemedicine has the potential to transform healthcare, it is important to address these security and privacy concerns to ensure that telemedicine is used in a responsible and effective manner [46].

Despite the benefits of using HIPAA and GDPR-compliant apps and software, their adoption remains limited [69]. To fully realize the potential of digital health and to advance global health goals, such as universal health coverage, a comprehensive data management framework that addresses the needs of real populations must be established at the national and international levels. Additionally, the advancement of international interoperability standards will improve the monitoring of health needs and the delivery of effective interventions [33].

Governments have a key role to play in enabling digital health innovation and addressing the privacy, accountability, and security of health data [70].

Similarly, another great concern is that of legacy liabilities. Telemedicine providers can be held liable for any medical errors or omissions that occur during virtual consultations, and it is important to ensure that they have the appropriate level of training and expertise, follow the same standards of care as in-person consultations, and have malpractice insurance [50,69]. According to Solimini et al., the use of telemedicine should also complement traditional healthcare services rather than replace them [62].

It is important to note that laws and regulations regarding telemedicine may vary by country, and it is important to be aware of the specific laws and regulations that apply in your jurisdiction. In addition, telemedicine raises important questions regarding ethical and legal issues, such as patient privacy and the security of health data, that must be addressed [44]. While telemedicine appears to provide many benefits, such as increased access to care and improved patient convenience, it is critical to address the security, ethical, and legal challenges that come with it as soon as possible in order to fully realize its potential and ensure its safe and effective implementation.

*Limitations*

This review has limitations, such as the exclusion of non-English articles, limited search sources (PubMed and Scopus), and no search sources for gray area literature.

By only focusing on articles that deal with digital health regulation, particularly data security, privacy, and legacy issues, the study may overlook other important aspects of the topic and may not provide a comprehensive view of the field. This narrow focus could also result in a skewed representation of the current state of research in this area.

## 5. Conclusions

Digital health has revolutionized the way medical services are delivered. The use of technology has made it possible for patients to receive advice and treatment remotely, ensuring their safety, especially during the COVID-19 pandemic. This has improved access to healthcare for people who may have difficulty traveling to see a doctor in person.

However, the digital health industry faces several challenges. One of the main challenges is the lack of standardized protocols for handling sensitive patient data that apply globally. This raises concerns about the confidentiality and security of patient information, especially in an era where data privacy is a major issue. Additionally, the absence of common legislative provisions regarding the exchange of confidential data makes it difficult to ensure that the delivery of effective care is maintained across different countries and regions.

To address these challenges, regulatory compliance is crucial. It will help clarify and harmonize all regulatory and normative aspects affecting digital health, making it possible to implement these services and make telemedicine mainstream. This will not only improve patient outcomes but also provide doctors with more certainty in their work, as they will know that they are following guidelines that have been established and agreed upon by regulatory authorities.

Another important consideration in the digital health landscape is the emergence of the informed "patient 4.0". This patient is often very well-informed about their health status and is not afraid to ask specific questions and challenge doctors, like a true expert. This can sometimes lead to defensive medicine, in which doctors, out of fear of making mistakes, could prescribe unnecessary tests and procedures. In an unregulated field such as digital health, this could be a major problem, and it is important to address it.

In conclusion, as a result of this scoping review, the authors determined that digital health has improved access to medical services. However, there are still major challenges that need to be addressed. Regulating the handling of patient data, clarifying legislative provisions, and addressing the challenges posed by informed patients are all critical steps in ensuring the successful implementation of telemedicine.

**Author Contributions:** Conceptualization, O.D.F. and G.L.M.; methodology, M.G.C.A.C. and G.L.M.; validation, G.C.; formal analysis, G.L.M.; investigation and data curation, O.D.F. and G.L.M.; writing— original draft preparation, O.D.F., M.G.C.A.C. and G.L.M.; writing—review and editing, O.D.F. and G.C.; supervision and project administration, G.C. All authors have read and agreed to the published version of the manuscript.

**Funding:** This research received no funding.

**Institutional Review Board Statement:** Not applicable.

**Informed Consent Statement:** Not applicable.

**Data Availability Statement:** Not applicable.

**Conflicts of Interest:** The authors declare that there is no conflict of interest.

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
