# Peer review of "Protection of Patient Data in Digital Oral and General Health Care: A Scoping Review with Respect to the Current Regulations"

_2673-6373, doi:10.3390/oral3020014_

Round 1

Reviewer 1 Report

- At a time when digital transformation is rapidly spreading, it is meaningful to review papers dealing with telemedicine issues in all areas of health care.

- The composition of the manuscript was well organized according to systematic procedures. However, revision of references is required after reviewing the regulations. Journal names are also a mixture of full names and abbreviations.

Author Response

1) At a time when digital transformation is rapidly spreading, it is meaningful to review papers dealing with telemedicine issues in all areas of health care.

  • Thank you for reviewing the article. Digital transformation is a complex and dynamic phenomenon that encompasses a wide range of applications and implications across various industries, including healthcare. As digital technologies continue to evolve and become more integrated into healthcare systems, it is becoming increasingly important to understand their impact on patient outcomes, healthcare delivery, and healthcare professionals' roles. As a result, some areas of healthcare may not have been fully investigated in the scoping review. Nonetheless, the eligibility criteria were essential in ensuring that the review's findings are specific and relevant to the research questions and objectives (as stated in the text: “We included studies conducted between January 2020 and December 2022, which focused solely on the use of mobile applications and software in healthcare. The review included all aspects of healthcare, including dental, nursing, and rehabilitation”). In this way, the scoping review provides a valuable overview of the current state of research on digital transformation in healthcare, highlighting the most significant areas of focus and potential avenues for future research. It is important to note that the purpose of this review is to provide a detailed overview of data protection, rather than covering telemedicine in general.

2) The composition of the manuscript was well organized according to systematic procedures. However, revision of references is required after reviewing the regulations. Journal names are also a mixture of full names and abbreviations.

  • The references have been modified according to editorial standards

Reviewer 2 Report

The authors aim to review the regulations about the use of digital health apps and software in healthcare. 

Introduction section must be improved with research gap and some statistical information.

Discuss data security and privacy policies in details with region-based examples in section 3.

Clearly write the analytical findings of table 2.

Authors should discuss about different regulatory compliances.

Author Response

The authors aim to review the regulations about the use of digital health apps and software in healthcare. 

  1. Introduction section must be improved with research gap and some statistical information.
  • Thank you for reviewing the article. The existing literature on data security and privacy in the context of digital health contains few region-specific details and cultural contexts. While there is a wealth of information on concepts and measures related to data security and privacy, there is a lack of in-depth analysis on how these measures are implemented and applied in different regions and cultural contexts. This lack of regional information is primarily due to the fact that data security and privacy laws and regulations vary widely across countries, making it difficult to provide a comprehensive global overview of these measures. Nevertheless, it's important to recognise these differences and understand how they affect the implementation and effectiveness of the measures. In addition, many studies on data security and privacy don't include concrete examples from different regions, which limits our understanding of how these measures are implemented in practise. We need more research to provide detailed analysis and case studies that highlight the challenges and best practises in implementing data security and privacy measures in different contexts. 

2. Discuss data security and privacy policies in details with region-based examples in section 3.

  • It is believed that the previous question has been answered

3. Clearly write the analytical findings of table 2.

  • The presentation of the results in table 2 has been enhanced for greater clarity and comprehensiveness.

4. Authors should discuss about different regulatory compliances.

  • The focus of this scoping review is to explore the current state of research on the Protection of patient data in digital oral and general health care. While regulatory compliance is an important aspect of this topic, it is not the primary objective of the review. Rather, the aim is to provide a comprehensive overview of the existing literature and identify any gaps in knowledge or areas that require further investigation. By taking a broad approach, this review improves and strengthens our understanding of how patient data protection can be effectively implemented in the digital healthcare context.

Reviewer 3 Report

This paper analyzes a number of papers on the current situation and issues of telemedicine regulation, and summarizes the key components of three health application regulations: privacy policies, data security policies, and professional medical-legal liabilities. The whole article, from determining the research problem, selecting the research content, drawing the data chart, to the final summary, has a clear structure. In the screening phase of the paper, the isolation and combination of the search terms is more comprehensive. It was then further reviewed and a table describing the detailed screening scope. The authors comb through existing research findings and give suggestions to improve and optimize digital health. The research data of this article is rich and has a wide range of research. According to these data, the authors have carried out detailed classification and statistics, which can see the rigor and logic of the authors in the research.

Minor comments:

-            In section 5, "patient 4.0" mentioned in the abstract is described only in the conclusion, very briefly, but not mentioned in the text, and it is suggested that the arrangement of this section should be reconsider.

-            In section 1 and section 4, after citing references, it is obviously that there is a lack of punctuation.

-            In Section 3, the explanation for “Identification of studies via databases and registers” is only “ After conducting a systematic search of multiple databases, we included 24 articles (Table 2) out of 290...... ”. Will a more detailed description of the process and data be more rigorous? The authors can think about it more.

Author Response

1. In section 5, "patient 4.0" mentioned in the abstract is described only in the conclusion, very briefly, but not mentioned in the text, and it is suggested that the arrangement of this section should be reconsider.

  • Thank you for reviewing the article. "Patient 4.0" has been removed from the abstract.

2. In section 1 and section 4, after citing references, it is obviously that there is a lack of punctuation.

  • The punctuation has been corrected.

3. In Section 3, the explanation for “Identification of studies via databases and registers” is only “ After conducting a systematic search of multiple databases, we included 24 articles (Table 2) out of 290...... ”. Will a more detailed description of the process and data be more rigorous? The authors can think about it more.

  • The presentation of the results has been enhanced for greater clarity and comprehensiveness.

Round 2

Reviewer 2 Report

The Authors have addressed all of my concerns with the original manuscript. The revised manuscript is ready for publication.